# Dataset Evaluation Method and Application for Performance Testing of SSVEP-BCI Decoding Algorithm

**DOI:** 10.3390/s23146310

**Published:** 2023-07-11

**Authors:** Liyan Liang, Qian Zhang, Jie Zhou, Wenyu Li, Xiaorong Gao

**Affiliations:** 1China Academy of Information and Communications Technology, Beijing 100161, China; 18618488256@163.com (L.L.);; 2Department of Biomedical Engineering, School of Medicine, Tsinghua University, Beijing 100084, China

**Keywords:** steady-state visual evoked potential (SSVEP), brain–computer interface (BCI), dataset, algorithm, performance testing

## Abstract

Steady-state visual evoked potential (SSVEP)-based brain–computer interface (BCI) systems have been extensively researched over the past two decades, and multiple sets of standard datasets have been published and widely used. However, there are differences in sample distribution and collection equipment across different datasets, and there is a lack of a unified evaluation method. Most new SSVEP decoding algorithms are tested based on self-collected data or offline performance verification using one or two previous datasets, which can lead to performance differences when used in actual application scenarios. To address these issues, this paper proposed a SSVEP dataset evaluation method and analyzed six datasets with frequency and phase modulation paradigms to form an SSVEP algorithm evaluation dataset system. Finally, based on the above datasets, performance tests were carried out on the four existing SSVEP decoding algorithms. The findings reveal that the performance of the same algorithm varies significantly when tested on diverse datasets. Substantial performance variations were observed among subjects, ranging from the best-performing to the worst-performing. The above results demonstrate that the SSVEP dataset evaluation method can integrate six datasets to form a SSVEP algorithm performance testing dataset system. This system can test and verify the SSVEP decoding algorithm from different perspectives such as different subjects, different environments, and different equipment, which is helpful for the research of new SSVEP decoding algorithms and has significant reference value for other BCI application fields.

## 1. Introduction

Brain–computer interface is a direct interaction, communication, and control system established between the brain and external devices without relying on peripheral nerves and muscle tissue [1,2,3]. The SSVEP paradigm is a classic brain–computer interface paradigm that has been extensively studied for over 20 years [4,5,6,7] due to its high signal-to-noise ratio, stable response, and high information transfer rate (ITR). Decoding algorithms are a crucial component of brain–computer interface research, as they process and analyze brain signals, and convert them into instructions that can be understood by external devices. Data required for algorithm research are typically self-collected by researchers recruiting subjects or obtained from public datasets published in the field. Thus, the publication of public datasets with standardized collection specifications and detailed descriptions is crucial for effectively verifying algorithm performance and promoting iterative algorithm progress. In recent years, several classic datasets in the SSVEP-BCI field have been released [8,9,10,11,12,13], and have been widely applied and verified, which effectively promoting the development of the SSVEP-BCI field.

However, the majority of current algorithmic research utilizes one or two datasets to verify their performance [14,15,16,17,18,19,20,21,22,23,24,25,26,27,28,29], which did not make full use of public data resources, and the results were limited by the distribution of data samples in individual datasets, so it was not conducive to judge the application effect of the algorithm in the actual scene through the result. This issue has two underlying causes. Firstly, existing datasets in the field lack unified organization and evaluation, with some datasets yet to gain widespread attention. Secondly, there is a lack of unified analysis and evaluation methods for multiple datasets in the field.

In order to solve the above problems, this paper sorted out six sets of frequency and phase modulation paradigms datasets [8,9,10,11,12,30] that were publicly available in the field of SSVEP-BCI. This paper further proposed an SSVEP dataset evaluation method and analyzed the above six datasets, forming a dataset system for algorithm performance testing that includes datasets with different sample population distributions, devices, and acquisition environments (as shown in Figure 1). Based on this dataset system, comprehensive testing of the SSVEP-BCI decoding algorithm performance was conducted, ultimately achieving an efficient simulation of the actual application effect of the decoding algorithm and avoiding significant differences between the actual BCI system application performance and the data validation results caused by independent dataset validation.

## 2. Materials and Methods

### 2.1. Datasets

The current research on decoding algorithms in the BCI field often corresponds to the paradigm. To ensure horizontal comparability between the dataset and the decoding algorithm, this paper selected the most influential frequency and phase modulation paradigm in the SSVEP field [31] (the existing public SSVEP datasets mainly adopt this paradigm), and then sorted out all the six public datasets of the frequency and phase modulation paradigm that can be obtained currently, as shown in Table 1.

SSVEP benchmark dataset (dataset1). The SSVEP benchmark dataset was collected and published by Wang et al. [8], containing EEG data from 35 subjects performing a 40-target frequency and phase modulation SSVEP typing task. Each stimulus target includes six trials, with each trial lasting 5 s. For more detailed information about the dataset1, please refer to reference [8]. This dataset will be referred to as dataset1 in this paper.

SSVEP BETA dataset (dataset2). The SSVEP BETA dataset [9] serves as an extension to the SSVEP benchmark dataset, with a different environment and subject group. Liu et al. [9] collected and released this dataset, which includes 70 participants and 4 online experiments. Each experiment consists of 40 trials, beginning with a 0.5 s cue period, followed by the flicker task period, and ending with a 0.5 s rest period. For the first 15 participants (S1–S15) in the dataset, the flicker task period lasts at least 2 s, while for the remaining 55 participants (S16–S70), it lasts at least 3 s. For more detailed information about dataset2, please refer to reference [9]. This dataset will be referred to as dataset2 in this paper.

SSVEP Wearable dataset (dataset3). The SSVEP wearable dataset [10] is currently the largest and most standardized wearable BCI dataset available within the SSVEP-BCI field. It contains experimental data from 102 participants, which was collected and released by Zhu et al. [10].

This dataset includes two collection methods of dry electrode and wet electrode, and each electrode collects 10 blocks of data, respectively. Each block consists of 10 trials per stimulus target, with a total duration of 2.84 s per trial, comprising a 0.5 s cue period, a 2 s task period, as well as an additional 0.14 s visual latency, and a 0.2 s visual after-effect period. For more detailed information on this dataset, please refer to the original publication [10]. Since this dataset includes data from both dry and wet electrodes, we will refer to them as dataset3a (dry electrode data) and dataset3b (wet electrode data), respectively in the following sections.

SSVEP elder dataset (dataset4). The SSVEP elder dataset [11] contains experimental data from 100 elderly participants and was collected and released by Liu et al. [11]. This dataset employs a nine targets SSVEP paradigm and is divided into six blocks. Each block consists of one trial for each stimulus target, with a total duration of 6 s per trial, including a 0.5 s pre-stimulus period, 5 s of stimulus presentation, and a 0.5 s post-stimulus period. For more detailed information on this dataset, please refer to the original publication [11]. This dataset will be referred to as dataset4 in this paper.

SSVEP FBCCA-DW dataset (dataset5). The FBCCA-DW dataset was collected by Yang et al. [30], including 14 subjects performing a 40-target SSVEP paradigm (same as dataset1). The experiment was divided into four blocks, with each block consisting of one trial for each stimulus target, and each trial lasting for 3 s. For more detailed information on this dataset, please refer to the original publication [30]. This dataset will be referred to as dataset5 in this paper.

SSVEP USCD dataset (dataset6). The USCD dataset was collected and published by Nakanishi et al. [12] and includes data from 10 healthy subjects. The dataset includes 12 stimulus targets, divided into 15 blocks with each block collecting one trial of each target. Each trial had a stimulus duration of 4 s. A detailed introduction to the dataset can be found in reference [12]. This dataset will be referred to as dataset6 in this paper.

### 2.2. Decoding Algorithm

This study tested the popular four decoding algorithms in the SSVEP-BCI field including canonical correlation analysis (CCA) [4], filter bank canonical correlation analysis (FBCCA) [32], ensemble task-related component analysis (eTRCA) [7], and task discriminant component analysis (TDCA) [33]. CCA, FBCCA, and eTRCA have all been considered as significant milestones in the development of SSVEP-BCI decoding algorithms and have led to many subsequent improvements. This study proposed a dataset system for evaluating SSVEP decoding algorithms and, therefore, focuses on the three most influential algorithms mentioned above. Furthermore, since the previously mentioned three algorithms were proposed several years ago (with the latest eTRCA algorithm being proposed in 2017), this study introduced a newly proposed eTRCA algorithm improvement called TDCA for validation. TDCA is also a widely recognized training-based decoding algorithm in the field.

Furthermore, among the four aforementioned algorithms, CCA and FBCCA are both non-training algorithms, while eTRCA and TDCA are training algorithms. In this study, eTRCA and TDCA algorithms were trained on data consisting of six trials for each task target (stimulation frequency) using cross-validation for trial selection. A filter bank consisting of five filters was used, and its design and weight curves were kept consistent with the reference [7]. In addition, the number of components in the TDCA algorithm was selected as nine after tuning (increasing it further did not improve performance but only added to the algorithm’s complexity). Another important parameter of TDCA is *Np*, which refers to the length of the sliding window. Typically, the data window length used for calculation needs to be greater than 1 s, so that *Np* can range from 0.1 s to 1 s (in intervals of 0.1 s). In this study, when the data window length was less than 1 s, the corresponding *Np* parameter was set as the maximum value supported by that window length. For instance, if the window length was 0.5 s, *Np* would take values from 0.1 s to 0.5 s (in intervals of 0.1 s). For more detailed information regarding the principles of the four decoding algorithms mentioned above, please refer to the respective references.

### 2.3. Dataset Evaluation Method

#### 2.3.1. Evaluation Indexes

The core feature of SSVEP is the EEG frequency response corresponding to the stimulation frequency [34]. Therefore, the frequency signal-to-noise ratio is the most direct indicator for evaluating the strength of the SSVEP-induced response, and it has two analysis methods: narrowband SNR and broadband SNR. In addition, the accuracy of recognition, optimal response time, and ITR are the core indicators of a broad consensus on the performance evaluation of the SSVEP-BCI algorithm [7,32]. The six analyzed datasets in this study possess varying paradigm coding parameters, collection equipment, and test groups. Consequently, the results of the aforementioned five indexes typically vary significantly among different samples [9]. To summarize, this paper proposed five indexes for data evaluation, including the narrow-band signal-to-noise ratio (SNR), wide-band SNR, accuracy of recognition, optimal response time, and information transfer rate (ITR).

In addition, since different algorithms yield different results for accuracy of recognition, optimal response time, and ITR, this study needs to select a widely accepted and influential decoding algorithm in the field as a standard algorithm. The FBCCA algorithm [32] is a milestone in the SSVEP-BCI field for its non-training decoding algorithm, which first introduced the concept of spatial filter banks and greatly improved the effectiveness of SSVEP decoding. Therefore, FBCCA is widely used in the development of BCI systems and is considered a highly influential and stable algorithm. Thus, the FBCCA was used as the standard algorithm for data analysis in this study.

The following section provides the specific definitions and calculation methods for the five indexes:
(1)The narrow-band SNR

The narrow-band SNR is defined as the ratio of the spectral amplitude of the stimulation frequency to its surrounding frequencies and is represented as *SNRt*. The calculation method is shown in Equation (1).
(1)$$SNRt=20log_{10}\frac{y\left(f\right)}{\sum_{k=1}^{1/df}\left[y\left(f-df\times k\right)+y\left(f+df\times k\right)\right]}.$$

Here, $y\left(f\right)$ refers to the signal amplitude spectrum calculated by fast Fourier transform, and *df* represents the spectral resolution. The *SNRt* in this context is defined as the ratio of the amplitude spectrum value of $y\left(f\right)$ to that of the frequencies 1 Hz apart on either side, and the unit is decibel (dB). During the calculation process, EEG data from the same frequency trials (usually 4–6 trials, depending on the data collection of different datasets) were first overlaid and averaged before calculating *SNRt*.
(2)Wide-band SNR

The wide-band SNR is defined as the ratio of the spectral amplitude of the stimulation frequency and its harmonic responses to the amplitude of other frequencies in a wideband range. This variable is represented as *SNRw* (as shown in Equation (2)).
(2)$$SNRw=20log_{10}\frac{\sum_{k=1}^{Nh}P\left(k\times f\right)}{\sum_{f=1}^{f=f_{s}/2}P\left(f\right)-\sum_{k=1}^{k=Nh}P\left(k\times f\right)},$$

*Nh* denotes the number of harmonics, *P*(*f*) refers to the power spectrum at frequency *f*, and *fs*/2 represents the Nyquist frequency.
(3)Accuracy of recognition using standard algorithm

The accuracy of multi-target recognition under FBCCA is referred to as *ACC_stand_*. The data window length used by the FBCCA algorithm is 2 s. If the window length of the dataset is less than 2 s, the maximum window length of the dataset is taken.
(4)Optimal response time of standard algorithm

Response time is an important index of BCI performance that can be matched to standard algorithms in the field for data analysis and discrimination with specific paradigms and scenarios. In this paper, the optimal response time for the multiple target recognition of data calculated by FBCCA is represented by *T_best_* (the time coordinate at the inflection points where the recognition accuracy is greater than 90%). If the maximum data window length has not yet reached an average recognition accuracy of 90%, then the optimal response time will be adjusted based on the accuracy obtained under the condition of the maximum data window length in the dataset. The specific calculation method is shown in Equation (3).
(3)$$T_{best}=90\%\times T_{max}/ACC_{T_{max}},$$
where *T_max_* represents the maximum data window length supported by the dataset, and $ACC_{T_{max}}$ represents the average recognition accuracy of the FBCCA algorithm under the condition of *T_max_* data window length.
(5)Optimal information transfer rate of standard algorithm

Information transfer rate (ITR) is an evaluation index that comprehensively considers recognition accuracy, coding target number, and response time. The calculation of ITR is shown in Equation (4):
(4)$$ITR=\frac{60}{T}\left\{\log_{2}N+P\log_{2}P+\left(1-P\right)\log_{2}\left[\frac{\left(1-P\right)}{\left(N-1\right)}\right]\right\},$$
*T* represents the calculation response window length, *N* represents the number of stimulation targets, and *P* represents the recognition accuracy. *T* is an important influencing factor in the calculation of ITR. When calculating ITR data in this research, the response time window length *T* was calculated as the recognition window length plus 0.5 s. This additional 0.5 s was considered to be the switching time that occurred when users shifted their gaze between targets in actual use scenarios [7,31]. Recognition accuracy is the most critical measure of BCI system performance as it determines the system usability. Therefore, the $ITR_{best}$ in this research is defined as the highest ITR score obtained under a response window length *T* corresponding to recognition rates exceeding 90%. If using any length of time window cannot achieve a recognition accuracy of over 90%, the calculation of the $ITR_{best}$ is based on the maximum time window length supported by the dataset.

#### 2.3.2. Index Scoring Method

Regarding the five evaluation indexes (*SNRt*, *SNRw*, *ACC_stand_*, *T_best_*, *ITR_best_*) introduced in the previous section, this study designed a percentage-based scoring method. The corresponding scoring calculation methods (*score*1–*score*5) were provided, and the total score was obtained by summing them up. However, *score*1–*score*5 were not designed to be 20 points for each item (the average of five indicators in percentage), but were slightly adjusted based on the importance of each indicator. Specifically, the corresponding total score of *SNRw* and *T_best_* were reduced to 15 points, while that of *ACC_stand_* and *ITR_best_* were increased to 25 points. The reasons for the specific adjustments are as follows: Although SNRw is an indicator that measures the strength of SSVEP response, the wideband noise can be reduced by filtering in the frequency or spatial domain during algorithm development (while narrowband noise is difficult to filter out). Therefore, *score*1 is not adjusted, while *score*2 is slightly reduced. Although *T_best_* is a crucial indicator that reflects the performance of the BCI system, the evaluation of the BCI system currently focuses on the accuracy and the information transfer rate [7,18,35]. Therefore, *score*4 was appropriately reduced, while *score*3 and *score*5 were appropriately increased.

In addition, this study introduced the log10 function to adjust the score change curve, aiming to compress the parameter range of the lower scores appropriately so that more complex data under different conditions can obtain some scores (e.g., dataset3b). The subsequent section will outline the specific calculation methods of *score*1–*score*5.
(1)Score of *SNRt*

The response *SNRt* distribution of SSVEP in the occipital region is typically in the range of −5 dB to 20 dB [9], thus a corresponding data score ‘*score*1’ can be given as shown in Equation (5).
(5)$$\begin{array}{cc} score1=0 & SNRt\leq -10\;\mathrm{dB}\\ score1=15.1log_{10}\left(SNRt+11\right) & SNRt\in \left(-10,\;10\right)\;\mathrm{dB}\\ score1=20 & SNRt\geq 10\;\mathrm{dB}. \end{array}$$

The value 15.1 is a parameter that ensures that *score*1 approximates 20 when *SNRt* reaches 10 dB, and it does not hold any physical significance. Similar parameter settings can be found in Equations (6)–(9) in the subsequent section.
(2)Score of *SNRw*

The response *SNRw* distribution of SSVEP in the occipital region is typically in the range of −35 dB to 0 dB [9], thus a corresponding data score ‘*score*2’ can be given as shown in Equation (6).
(6)$$\begin{array}{cc} score2=0 & SNRw\leq -40\;\mathrm{dB}\\ score2=10log_{10}\left(SNRt+41\right) & SNRw\in \left(-40,\;-10\right)\;\mathrm{dB}\\ score2=15 & SNRw\geq 10\;\mathrm{dB}. \end{array}$$
(3)Score of the *ACC_stand_*

Using the FBCCA algorithm [32] to compute the data, the accuracy of multiple target identification is referred to as *ACC_stand_*. In consideration of system performance and availability, a corresponding score can be assigned to represent data evaluation, denoted as *score*3 and shown in Equation (7).
(7)$$\begin{array}{cc} score3=0 & ACC_{stand}\leq 50\%\\ score3=15.5log_{10}\left(ACC_{stand}-49\right) & ACC_{stand}\in \left(50,\;90\right)\%\\ score3=25 & ACC_{stand}\geq 90\% \end{array}$$
(4)Score of the *T_best_*

The calculation method of *score*4 (score of *T_best_*) is presented in Equation (8). If the recognition accuracy can achieve 90% within 2 s data length, full marks will be awarded for this item. If the *T_best_* falls between 2 and 8 s, a corresponding score will be assigned based on the number of targets (the *C* in Equation (8)) and *T_best_*. If *T_best_* exceeds 8 s, indicating low system availability, the score for this item will be zero.
(8)$$\begin{array}{cc} score4=15 & T\leq 2\;s\\ score4=19-21.5log_{10}\left(T\right)+log_{10}\left(C\right) & T\in \left(2,\;8\right)s,C\in \left(1,\;160\right)\\ score4=0 & T\geq 8\;s \end{array}$$
(5)Score of the *ITR_best_*

When using the FBCCA algorithm to compute the classic dataset in this paper, the obtained *ITR_best_* falls within the range of 30 bits/min to 100 bits/min. Hence, the score calculation for *ITR_best_* (*score*5) is given by Equation (9).
(9)$$\begin{array}{cc} score5=0 & ITR_{best}<30\;\mathrm{bits}/\min\\ score5=14g_{10}\left(ITR_{best}-29\right) & 30\;\mathrm{bits}/\min<ITR_{best}\leq 100\;\mathrm{bits}/\min\\ score5=25 & 100\;\mathrm{bits}/\min<ITR_{best}. \end{array}$$
(6)Total score

This study established the data range of relevant indexes and provided corresponding score assessments based on the analysis and judgment of SSVEP datasets. By summing up the scores of five indexes and referring to Table 2, the level of SSVEP datasets can be determined. Table 2 shows that the higher the final score, the lower the decoding difficulty of the dataset.

### 2.4. Algorithm Performance Evaluation Index

To evaluate algorithm performance, this study used recognition accuracy, best response time, and ITR as indices. The definitions and computation methods of these indices have been introduced in the dataset evaluation method section.

## 3. Result

### 3.1. The Calculation Results of Six Datasets

Table 3 presents the calculation results for the six datasets based on the five SSVEP dataset evaluation indexes and six score (*score*1–*score*5 and total score) proposed in this paper.

Table 3 indicates that there are differences in the evaluation scores of six datasets in terms of *SNRt*, *SNRw*, *ACC_stand_*, *T_best_*, and *ITR_best_*. Dataset3a, which uses wearable SSVEP data collected by dry electrode headbands has the highest decoding difficulty, while dataset1, which strictly controls subject age, state, and collection environment has the lowest decoding difficulty due to its outstanding decoding performance. To summarize, the aforementioned six datasets establish a system for verifying algorithm performance with a vertically distributed decoding difficulty rating.

### 3.2. Algorithm Performance Testing

Comprehensive testing and verification of the performance of CCA, FBCCA, eTRCA, and TDCA algorithms were conducted using the six datasets analyzed in this paper, with the results shown in Figure 2, Table 4 and Table 5. Figure 2 examines the performance of the same algorithm on different datasets, while Table 4 and Table 5 examine the performance of different algorithms on the same dataset.

In Figure 2, the data lengths refer to the duration of the extracted EEG data starting from the onset of the stimulus trial, representing the length of EEG data used for algorithm recognition. From Figure 2, it can be observed that the same algorithm exhibited differences in recognition accuracy when applied to different datasets. Furthermore, these differences vary based on different data window length conditions. For instance, in the case of the CCA algorithm, the recognition result for dataset1 is the lowest (among the six datasets) with a 0.4-s time window length. However, when the window length is extended to 1 s or longer, there is a significant improvement in the recognition result for dataset1. This also indicates that the six datasets mentioned above do exhibit a vertical distribution decoding difficulty and can be comprehensively tested and evaluated for algorithm performance in different subject populations, collection devices, and collection environments.

Using a window length of 1 s, a comparison study between any two methods was conducted using the paired sample *t*-test. The corresponding *p*-value after the Bonferroni correction is listed in Table 5.

Table 4 and Table 5 demonstrate that the accuracy calculation results of eTRCA and TDCA algorithms are higher than those of CCA and FBCCA algorithms, and the performance of TDCA is better than that of eTRCA, while the performance of FBCCA is better than that of CCA. However, there are exceptions, such as when using dataset3 with high decoding difficulty, the computational performance of FBCCA and CCA is similar. When using dataset6, the computational performance of TDCA and eTRCA is comparable. These results indicate that the performance of algorithms under different environmental and demographic conditions is not fixed.

Furthermore, after Bonferroni correction, there were no significant differences in the calculation results between eTRCA and TDCA algorithms. However, the independent T-tests between the two algorithms showed significant differences in some datasets (Dataset1: *p* < 0.05; Dataset2: *p* < 0.001; Dataseta3a: *p* < 0.001; Dataset3b: *p* < 0.001; Dataset4: *p* < 0.001; Dataset5: *p* = 0.09; Dataset6: *p* = 0.67). Moreover, although the statistical results calculated for dataset5 after Bonferroni correction were non-significant, the T-test results between these four algorithms without Bonferroni correction showed significant differences (except for the *p*-value between eTRCA and TDCA).

Additionally, the decoding performance of specific datasets is not fixed. Taking dataset1 as an example, it can be observed that the decoding performance of this dataset is significantly worse when using non-training algorithms (CCA and FBCCA) compared to dataset4 and dataset5. However, the opposite results were obtained when using training algorithms (eTRCA and TDCA). These results suggest that different algorithms perform differently on the gradients of the six datasets presented in terms of decoding difficulty. Therefore, the proposed decoding difficulty gradient distribution in this paper may facilitate comprehensive performance testing of different algorithms.

### 3.3. Result of Typical Subjects

There are large individual differences in the brain response between different people. Therefore, when analyzing the performance of the decoding algorithm, we should not only focus on the mean value of the calculated results but also pay attention to the lowest and highest values. The highest value indicates the performance upper limit that the algorithm can achieve. For example, if the corresponding BCI system is applied in specific operational scenarios [36,37], selecting individuals with excellent performance from a large pool of operators can achieve the best system performance. The minimum value indicates the lower limit of algorithmic performance, which should be improved in mass usage scenarios [3,38,39] to enable a wider range of users to use it. Table 6 shows the statistical results of the best and worst subjects for the four algorithms analyzed on six datasets under the condition of a 2 s data window length.

Table 6 shows that there is a large difference in accuracy and ITR between the best and worst performance subjects for the same algorithm and dataset. Additionally, for the same dataset, the best and worst subjects for different algorithms are not entirely identical. This result objectively reflects the individual variability of SSVEP response. Furthermore, the worst performing subjects did not achieve an identification accuracy above 80% for the 2 s data window length (except for eTRCA and TDCA on dataset1 and dataset6), while the accuracy of the best performance subjects was all exceeding 90%. These results suggest that further optimization and improvement are needed for the SSVEP-BCI developed based on the above four algorithms to enhance the adaptability of acquisition equipment and environment, as well as expand the applicable population.

## 4. Discussion

### 4.1. Advantages of the Dataset System over Individual Datasets

The dataset system proposed in this paper is valuable because it can verify the performance of SSVEP-BCI decoding algorithms under different conditions, including different subject populations, acquisition devices, acquisition environments, and decoding difficulties. Figure 3 summarizes the results from Figure 2, Table 4 and Table 5, indicating that the performance of the same decoding algorithm varies significantly across different datasets, and not all calculated results meet the performance requirements for BCI systems in actual scenarios.

Therefore, if only 1–2 datasets are used in the decoding algorithm research process, the algorithm’s performance may fluctuate significantly in actual application scenarios. It is recommended to use multiple datasets to test and evaluate decoding algorithms to ensure their reliability and effectiveness in various real-world scenarios.

Further analysis Table 4 and Table 5, taking the most widely used dataset (dataset1) as an example, shows that eTRCA achieves an identification accuracy of 94.3 ± 10.9% in a 1 s window length, while TDCA achieves an accuracy of 96.5 ± 6.1% and FBCCA achieves an identification accuracy of 90.2 ± 13.3% for a 2 s window length. Without further validation on other datasets, these accuracy results suggest that FBCCA, eTRCA, and TDCA are expected to meet the needs of actual BCI use.

However, in reality, the actual performance of these algorithms can vary significantly when the acquisition environment, participant group, and electrode materials change. For instance, when using dataset2 to simulate real application environments with a more open acquisition environment and a wider range of participant groups, eTRCA achieves only 75.5 ± 22.0% accuracy with a 1 s window length, while TDCA achieves an accuracy of 82.5 ± 16.6% with a 1 s window length, and FBCCA achieves an accuracy of 81.8 ± 15.7% with a 2 s window length. Moreover, when using dry electrode data in dataset3a, the accuracy of these algorithms will decrease to 58.1 ± 31.0% (eTRCA using 1 s window length), 79.4 ± 21.7 (TDCA using 1-s window length), and 59.3 ± 27.3% (FBCCA using a 2 s window length). These identification accuracy results are difficult to meet the practical requirements of BCI, and the difference between them and the high-performance results obtained under strictly controlled conditions (dataset1) is significant. Thus, it is highly recommended to use multiple datasets with a gradient distribution of the decoding difficulty level to test the performance of decoding algorithms in SSVEP-BCI.

Furthermore, the dataset system constructed in this study can be expanded in the future to include more high-value datasets that reflect realistic BCI scenarios, as shown in Figure 3. This will enhance the accuracy and applicability of decoding algorithms in practical BCI systems.

### 4.2. Value of Dataset Decoding Difficulty Assessment

This study evaluated six datasets of the frequency and phase modulation SSVEP paradigm (as shown in Table 3). The six datasets covered decoding difficulty ratings of A, B, C, D, and E, and the rating results corresponded to the results in Figure 2 and Table 4, indicating that the performance indexes calculated by different decoding algorithms were relatively lower for datasets with higher decoding difficulty. Therefore, the dataset decoding difficulty assessment method can be used to construct a dataset system with differentiated decoding difficulty levels to achieve comprehensive testing of decoding algorithm performance. Furthermore, it can provide a reference for algorithm performance testing and improvement. For example, although CCA, FBCCA, eTRCA, and TDCA performed poorly on dataset3a, the decoding difficulty rating of E for this dataset confirms the rationality of the results and provides ideas for subsequent improvements to the decoding algorithm (i.e., studying the signal characteristics of dataset3a and focusing on improving algorithm performance based on dataset3a).

### 4.3. Current Situation of Dataset Usage in SSVEP Decoding Algorithm Research

This study conducted a further analysis of the current state of dataset usage in the latest research papers on SSVEP decoding algorithms. The selection of papers followed specific criteria, including using the Web of Science Core Collection with the keywords “SSVEP & algorithm” as the primary topic, limiting the document type to papers, and restricting the search results to papers published before April 2023. In addition, we manually excluded studies that were not related to decoding algorithms such as research on BCI system applications involving algorithm validation and usage, algorithm review papers, and dataset papers. Subsequently, we sorted the search results chronologically and selected the 20 most recently published papers and analysis of their dataset usage, which is presented in Table 7.

Table 7 shows that the majority of the 20 SSVEP decoding algorithm papers analyzed used more than one dataset, with only three papers [18,20,25] using self-collected data verification methods. Among the 20 papers, 40% (8/20) used only one dataset, while 80% (16/20) used no more than two datasets, and the remaining four papers used only three datasets for verification. Therefore, there is significant room for expansion in dataset usage in current research on SSVEP decoding algorithms.

Regarding dataset usage, it can be seen that in the 20 papers of Table 7, SSVEP Benchmark [8] (dataset1) was used 11 times, SSVEP BETA [9] (dataset2) was used 5 times, SSVEP UCSD [12] (dataset6) was used 4 times, and SSVEP Wearable [10] (dataset3) was used twice (this dataset was released in 2021). Other datasets were used no more than twice. It can be seen that there are significant differences in the frequency of use of different datasets. From the perspective of the number of subjects, it can be observed that the use of previous datasets for algorithm verification involves a much larger number of subjects than using self-collected data. Therefore, releasing SSVEP datasets can promote algorithm research. It is hoped that research teams with capabilities in this field can collect more self-generated validation data and make them public on the basis of using existing datasets.

In summary, the dataset evaluation method and algorithm testing dataset system proposed in this study can provide more references for future algorithm research. The decoding algorithm research can attempt performance testing and verification on more newly shared datasets (such as dataset2, dataset3, and dataset4).

### 4.4. Subsequent Extensions of This Study

This paper proposed a decoding difficulty evaluation method for SSVEP datasets. Among them, *ACC_stand_*, *T_best_*, and *ITR_best_* are all based on the calculation results of the most classic non-training algorithm FBCCA. However, during the actual analysis process (Figure 2, Table 4 and Table 5), it can be found that the *ACC_stand_*, *T_best_*, and *ITR_best_* of different decoding algorithms are different. Therefore, in future studies, a variety of decoding algorithms can be explored and used to comprehensively evaluate the decoding difficulty of the datasets.

Furthermore, this study analyzed four representative SSVEP decoding algorithms that have the potential for further testing and expansion to include other algorithms. Additionally, there is still room for expansion of the six frequency and phase modulation SSVEP datasets compiled in this study. Future studies can attempt to expand the analysis to include datasets from SSVEP paradigms that are not entirely identical (For example, combining the frequency and phase modulation paradigm with the dual-frequency modulation SSVEP paradigm [42]). This can promote the development of performance improvements for SSVEP-BCI general decoding algorithms (e.g., eTRCA can be used for various SSVEP paradigms). Further attempts can also be made to conduct comprehensive analyses of visual BCI paradigms that are similar to SSVEP paradigms (such as the C-VEP paradigm [51]) to promote research on general decoding algorithms for visual BCI. Other BCI paradigms (such as the motor imagery paradigm [52], P300 paradigm [41], etc.) can also build corresponding dataset systems based on their own data characteristics.

## 5. Conclusions

In summary, this paper summarized six open-source datasets of the frequency and phase modulation paradigm of SSVEP-BCI. Additionally, this paper proposed a dataset decoding difficulty evaluation method and integrated the above six datasets to form a set of SSVEP algorithm performance testing dataset systems. Finally, based on the dataset system, the existing four classic algorithms (CCA, FBCCA, eTRCA, and TDCA) were tested and verified, and significant differences were found in the calculation results of different algorithms on different datasets. This demonstrates the importance of utilizing an algorithm evaluation dataset system that includes different subject groups, scenarios, and collection devices for new SSVEP decoding algorithm research. Furthermore, this work may serve as a reference for dataset evaluation and algorithm performance testing of other BCI paradigms.

## Figures and Tables

**Figure 1 sensors-23-06310-f001:**
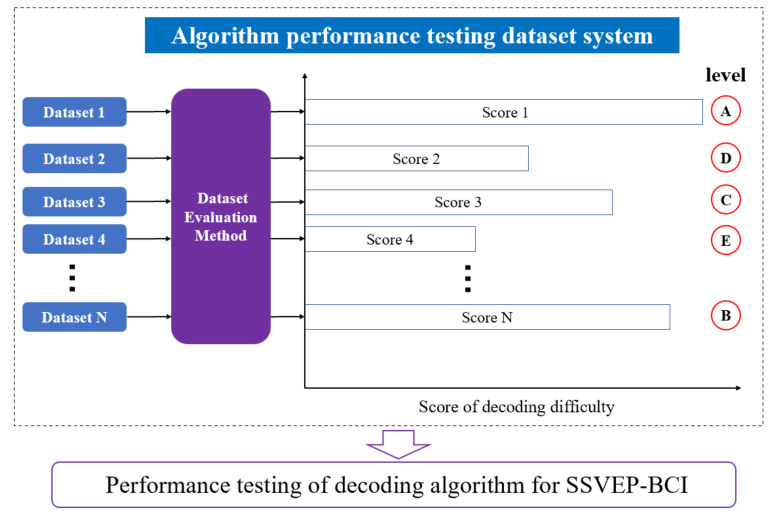
Dataset evaluation method and application for performance testing of SSVEP-BCI decoding algorithm.

**Figure 2 sensors-23-06310-f002:**
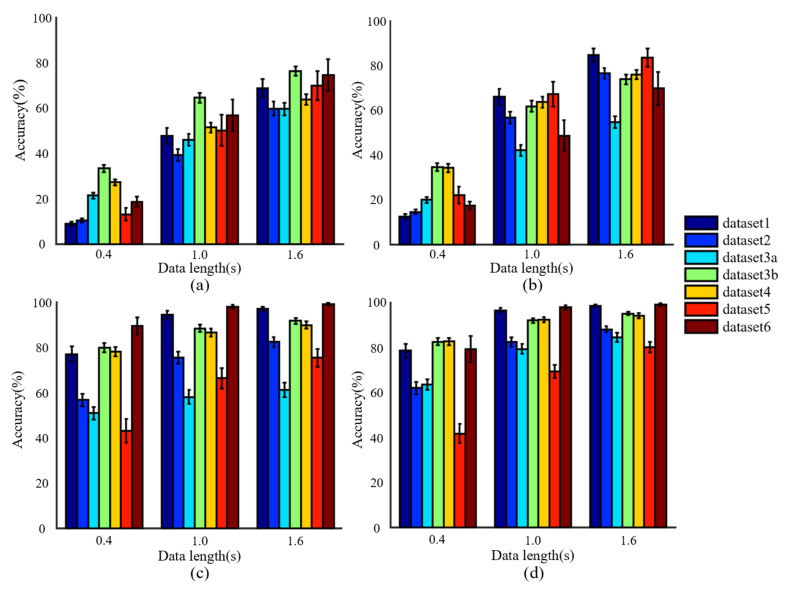
Recognition accuracy statistics for the same algorithm on different datasets. (**a**) CCA (**b**) FBCCA (**c**) eTRCA (**d**) TDCA.

**Figure 3 sensors-23-06310-f003:**
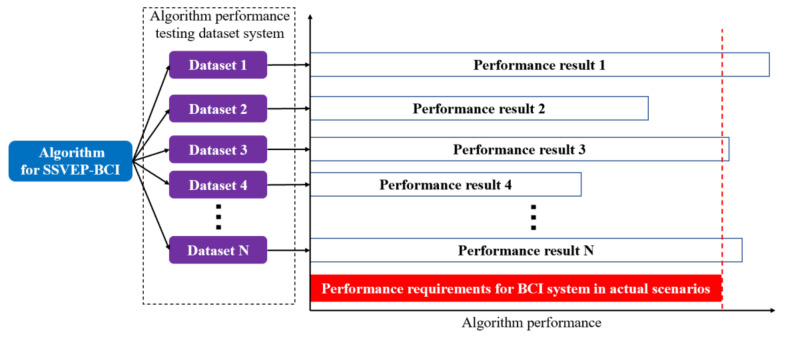
Performance testing SSVEP-BCI decoding algorithm based on dataset system.

**Table 1 sensors-23-06310-t001:** Information statistics of six SSVEP-BCI datasets.

Dataset	Citation	Subjects	Target	Channels
Benchmark dataset (dataset1)	Wang et al. [8]	35	40	64
SSVEP BETA dataset (dataset2)	Liu et al. [9]	70	40	64
SSVEP Wearable dataset (dataset3)	Zhu et al. [10]	102	12	8
SSVEP elder dataset (dataset4)	Liu et al. [11]	100	9	64
SSVEP FBCCA-DW dataset (dataset5)	Yang et al. [30]	14	40	64
SSVEP UCSD dataset (dataset6)	Nakanishi et al. [12]	10	12	8

**Table 2 sensors-23-06310-t002:** Decoding difficulty level of SSVEP dataset.

Total Score	Decoding Difficulty Level
[85, 100]	A
[70, 85)	B
[55, 70)	C
[40, 55)	D
[0, 40)	E

**Table 3 sensors-23-06310-t003:** Results of decoding difficulty evaluation for datasets.

Dataset	Dataset1	Dataset2	Dataset3a	Dataset3b	Dataset4	Dataset5	Dataset6
*SNRt* (dB)	5.1 ± 8.1	−3.1 ± 6.5	−9.6 ± 4.2	−9.9 ± 4.6	2.0 ± 8.7	−2.9 ± 6.6	4.0 ± 7.7
Score1	16.5	11.2	4.9	4.8	14.5	11.6	16.1
*SNRw* (dB)	−11.5 ± 6.1	−10.5 ± 5.1	−54.4 ± 7.9	−36.0 ± 8.4	−10.6 ± 5.1	−36.9 ± 9.7	−33.8 ± 14.3
Score2	14.4	14.6	0.4	5.2	14.6	5.6	6.8
*ACC_stand_* (%)	90.2 ± 13.5	81.8 ± 15.8	59.3 ± 27.4	77.9 ± 21.6	80.3 ± 19.7	89.4 ± 10.8	75.8 ± 23.0
Score3	23.4	21.6	12.4	19.0	20.5	24.0	18.6
*T_best_* (s)	1.9 ± 0.8	2.2 ± 0.8	4.1 ± 3.4	2.3 ± 1.5	3.2 ± 2.6	1.8 ± 0.7	3.0 ± 1.6
Score4	13.6	13.1	8.9	12.1	10.2	13.8	10.5
*ITR_best_* (bits/min)	126.6 ± 41.1	101.2 ± 40.5	40.1 ±37.3	65.8 ±41.3	60.4 ±40.4	124.7 ±44.4	54.3 ±28.4
Score5	24.4	22.6	9.7	16.1	14.6	24.5	16.5
Total score	92.4	83.2	36.3	57.1	74.4	79.5	68.5
Level	A	B	E	C	B	B	C

**Table 4 sensors-23-06310-t004:** Accuracy calculation results of four algorithms on different datasets.

Accuracy (%)					
Data length	Dataset\Algorithm	CCA	FBCCA	eTRCA	TDCA
0.5 s	Dataset1	14.2 ± 8.0	21.4 ± 12.0	81.9 ± 19.1	**84.6** ± **15.9**
	Dataset2	15.3 ± 10.1	21.9 ± 12.5	60.8 ± 23.5	66.4 ± 21.9
	Dataset3a	**27.3** ± **17.0**	**25.7** ± **17.1**	52.6 ± 29.1	**68.4** ± **23.4**
	Dataset3b	40.9 ± 19.1	43.0 ± 20.6	81.5 ± 20.2	**85.6** ± **14.4**
	Dataset4	34.1 ± 16.2	43.0 ± 19.6	81.3 ± 20.2	**86.9** ± **15.2**
	Dataset5	21.1 ± 13.6	31.1 ± 16.6	**48.0** ± **19.1**	47.8 ± 15.7
	Dataset6	**24.4** ± **9.8**	**23.8** ± **8.7**	**92.5** ± **8.7**	86.6 ± 14.2
1 s	Dataset\Algorithm	CCA	FBCCA	eTRCA	TDCA
	Dataset1	47.8 ± 20.5	66.0 ± 21.7	94.3 ± 10.9	**96.5** ± **6.1**
	Dataset2	39.4 ± 22.4	56.9 ± 22.0	75.5 ± 22.0	**82.5** ± **16.6**
	Dataset3a	**46.0** ± **25.7**	**42.1** ± **25.2**	58.1 ± 31.0	**79.4** ± **21.7**
	Dataset3b	**64.6** ± **23.0**	**61.9** ± **24.2**	88.4 ± 15.8	**92.0** ± **15.8**
	Dataset4	51.5 ± 22.2	63.7 ± 23.7	86.7 ± 18.0	**92.4** ± **11.8**
	Dataset5	50.3 ± 25.3	67.4 ± 20.8	66.3 ± 16.6	**69.4** ± **11.4**
	Dataset6	**56.9** ± **22.2**	**48.8** ± **21.5**	**98.1** ± **2.8**	97.8 ± 3.4
2 s	Dataset\Algorithm	CCA	FBCCA	eTRCA	TDCA
	Dataset1	77.6 ± 21.3	90.2 ± 13.3	98.3 ± 3.0	**99.0** ± **1.3**
	Dataset2	67.4 ± 24.2	81.8 ± 15.7	85.8 ± 15.3	**90.5** ± **9.5**
	Dataset3a	**63.2** ± **26.3**	**59.3** ± **27.3**	61.2 ± 32.1	**81.2** ± **22.6**
	Dataset3b	**80.3** ± **19.8**	**77.9** ± **21.5**	88.1 ± 19.3	**95.8** ± **6.7**
	Dataset4	67.7 ± 23.0	80.3 ± 19.6	91.2 ± 15.2	**95.2** ± **10.0**
	Dataset5	78.8 ± 20.6	89.4 ± 10.4	79.5 ± 13.1	**84.1** ± **7.1**
	Dataset6	**80.4** ± **20.4**	**75.8** ± **21.9**	99.4 ± 1.7	**99.4** ± **1.3**

**Table 5 sensors-23-06310-t005:** Accuracy comparison results based on the paired t-test with Bonferroni correction.

Method Comparison (M1 vs. M2)	Dataset1	Dataset2	Dataset3a	Dataset3b	Dataset4	Dataset5	Dataset6
CCA vs. FBCCA	< ***	< ***	N.S.	N.S.	< ***	N.S.	N.S.
CCA vs. eTRCA	< ***	< ***	< **	< ***	< ***	N.S.	< ***
CCA vs. TDCA	< ***	< ***	< ***	< ***	< ***	N.S.	< ***
FBCCA vs. eTRCA	< ***	< ***	< ***	< ***	< ***	N.S.	< ***
FBCCA vs. TDCA	< ***	< ***	< ***	< ***	< ***	N.S.	< ***
eTRCA vs. TDCA	N.S.	N.S.	N.S.	N.S.	N.S.	N.S.	N.S.

1: < *** and < ** denote that M1 is worse than M2 at different significant levels of 0.001 and 0.01. 2: N.S. indicates no significant difference between M1 and M2.

**Table 6 sensors-23-06310-t006:** The accuracy of the best performance subjects and the worst performance subjects.

Dataset	Algorithm	Worst Sub			Best Sub		
		Sub	2 s ACC (%)	2 s ITR (bits/min)	Sub	2 s ACC (%)	2 s ITR (bits/min)
Dataset1	CCA	s16	27.9	15.8	s5	98.3	122.7
	FBCCA	s11	52.5	43.5	s25	100.0	127.7
	eTRCA	s19	87.5	98.8	s14	100.0	127.7
	TDCA	s33	**93.3**	**110.8**	s13	100.0	127.7
Dataset2	CCA	s17	16.3	6.1	s23	100.0	127.7
	FBCCA	s41	37.5	25.5	s23	100.0	127.7
	eTRCA	s61	38.8	26.9	s23	99.4	125.6
	TDCA	s44	**65.0**	**60.9**	s30	100.0	127.7
Dataset3a	CCA	s74	**10.0**	**0.1**	s87	100.0	86.0
	FBCCA	s22	9.2	**0.0**	s52	100.0	86.0
	eTRCA	s96	4.2	**0.5**	s7	100.0	86.0
	TDCA	s74	5.8	**0.2**	s7	100.0	86.0
Dataset3b	CCA	s46	20.8	2.6	s7	100.0	86.0
	FBCCA	s42	20.8	2.6	s7	100.0	86.0
	eTRCA	s1	15.0	0.8	s3	100.0	86.0
	TDCA	s61	**70.0**	**40.0**	s3	100.0	86.0
Dataset4	CCA	s21	11.1	0.0	s31	100.0	76.1
	FBCCA	s34	15.9	0.4	s27	100.0	76.1
	eTRCA	s34	14.3	0.2	s3	100.0	76.1
	TDCA	s24	**42.9**	**11.3**	s3	100.0	76.1
Dataset5	CCA	s8	33.1	20.9	s1	99.4	125.6
	FBCCA	s8	63.8	59.1	s1	**100.0**	**127.7**
	eTRCA	s6	44.4	33.4	s1	93.1	110.3
	TDCA	s4	**70.0**	**68.5**	s1	93.8	111.7
Dataset6	CCA	s2	37.8	11.4	s8	100.0	86.0
	FBCCA	s2	35.6	10.0	s8	99.4	84.4
	eTRCA	s2	94.4	74.0	s1	100.0	86.0
	TDCA	s2	**95.6**	**76.1**	s1	100.0	86.0

**Table 7 sensors-23-06310-t007:** The current situation of dataset usage in SSVEP decoding algorithm research.

Citation	Data Sources	Data Publicly Available	Experiment	Subjects	Year
Mu et al. [14]	Mu et al. [40]	--	offline	9	2022
Oikonomou et al. [15]	EPOC dataset [13]	Yes	offline	11	2023
Wang et al. [16]	SSVEP Benchmark [8]	Yes	offline	35	2023
Guney et al. [17]	SSVEP Benchmark [8]	Yes	offline	35	2023
	SSVEP BETA [9]	Yes		70	
Zhang et al. [18]	Self-collected	--	offline	20	2023
Yin et al. [19]	SSVEP Benchmark [8]	Yes	offline	35	2023
Ke et al. [20]	Self-collected	--	offline online	15 14	2023
Wong et al. [35]	Chen et al. [41]	Yes	offline	8 and 12	2023
	Liang et al. [42]	Yes		12	
	Chang et al. [43]	--		12	
Tabanfar et al. [21]	Kołodziej et al. [44]	--	offline	5	2023
Lee et al. [22]	SSVEP Benchmark [8]	Yes	offline	35	2023
	OpenBMI dataset [45]	Yes		54	
Bian et al. [46]	SSVEP Benchmark [8]	Yes	offline	35	2023
	SSVEP BETA [9]	Yes		70	
	SSVEP UCSD [12]	Yes		10	
Ziafati et al. [23]	Zhang et al. [47]	Yes	offline	10	2023
Luo et al. [24]	SSVEP Benchmark [8]	Yes	offline	35	2022
	SSVEP BETA [9]	Yes		70	
Chuang et al. [25]	Self-collected	--	offline	24	2022
Pan et al. [26]	SSVEP UCSD [12]	Yes	offline	10	2022
	Wang et al. [48]	--		10	
Oikonomou et al. [27]	SSVEP Benchmark [8]	Yes	offline	35	2022
	EPOC dataset [13]	Yes		11	
Zhou et al. [49]	SSVEP UCSD [12]	Yes	offline	10	2022
	SSVEP Benchmark [8]	Yes		35	
	SSVEP BETA [9]	Yes		70	
Yan et al. [28]	SSVEP Benchmark [8]	Yes	offline	35	2022
	SSVEP UCSD [12]	Yes		10	
Zhang et al. [29]	SSVEP Benchmark [8]	Yes	offline	35	2022
	SSVEP Wearable [10]	Yes		102	
Bassi et al. [50]	SSVEP Benchmark [8]	Yes	offline	35	2022
	SSVEP BETA [9]	Yes		70	
	SSVEP Wearable [10]	Yes		102	

## Data Availability

The data that support the findings of this study are openly available. Among them, dataset1-dataset5 can be downloaded from “bci.med.tsinghua.edu.cn” (accessed on 9 January 2023) The download instructions for dataset6 can be found in the article written by Nakanishi et al. [12].

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
