# Peer review of "Dataset Evaluation Method and Application for Performance Testing of SSVEP-BCI Decoding Algorithm"

_sensors, 2023, doi:10.3390/s23146310_

Round 1

Reviewer 1 Report

The proposed problem is valuable for validating SSVEP-BCI methods. However, the selection of dataset, decoding algorithms, and evaluation indexes are in lack of strong motivation. Besides, quite a few mistakes reduced the quality of the paper. Sometime, the paper needs better manner to present.

1. The motivation to select the 6 dataset, the 4 decoding algorithms, and the 6 evaluation indexes are missing. Just because they are commonly cited or used? Do they have some charactors deserve attention? For instance, some charactors lead to the differences when applied to different datasets? Convincing reasons are needed to support the choices.

2. The criteria of the selection of 20 most cited papers and 20 latest papers are weak. Why these papers are selected? Do they have something different to others, or do they more strongly reflect the proposed problem(performance differences when different datasets are used)?

3. line 260, "Based on the five SSVEP dataset evaluation indexes proposed in this paper...", isn't it 6 indexes?

4. Why FBCCA is used as standard algorithm?

5. Lines 124-126, "This paper proposed six indexes for data evaluation, including the narrow-band sig-124 nal-to-noise ratio (SNR), wide-band SNR, accuracy of recognition using standard algo-125 rithm, optimal response time of standard algorithm, and information transfer rate of 126 standard algorithm." But it seems only 5 indexes.

6. Lines 290-306 are confusion. A better presentation should be considered.

Few spots need to be improved.

Author Response

Authors’ Responses

The authors thank the editor and the reviewer for their constructive comments and suggestions. We have carefully revised the manuscript according to the comments. Modifications in the manuscript have been marked in red font (also in the article). The detailed responses to the comments are as follows.

  1. The motivation to select the 6 dataset, the 4 decoding algorithms, and the 6 evaluation indexes are missing. Just because they are commonly cited or used? Do they have some charactors deserve attention? For instance, some charactors lead to the differences when applied to different datasets? Convincing reasons are needed to support the choices.

Response:

Thank you for your question and suggestions. Indeed, this article needs a more in-depth introduction and discussion about the selection of datasets, algorithms, and indicators. Relevant adjustments have been made to the manuscript as follows:

2.1 Datasets

The current research on decoding algorithms in the BCI field often corresponds to the paradigm. To ensure horizontal comparability between the dataset and the decoding algorithm, this paper selected the most influential frequency and phase modulation paradigm in the SSVEP field[31] (the existing public SSVEP datasets mainly adopt this paradigm), and then sorted out all the six public datasets of the frequency and phase modulation paradigm that can be obtained currently, as shown in Table 1.

2.2 Decoding algorithm

This study tested the popular four decoding algorithms in the SSVEP-BCI field, including canonical correlation analysis (CCA) [4], filter bank canonical correlation analysis (FBCCA) [32], ensemble task-related component analysis (eTRCA)[7], and task discriminant component analysis (TDCA) [33]. CCA, FBCCA, and eTRCA have all been considered as significant milestones in the development of SSVEP-BCI decoding algorithms, and have led to many subsequent improvements. This study proposed a dataset system for evaluating SSVEP decoding algorithms and therefore focuses on the three most influential algorithms mentioned above. Furthermore, since the previously mentioned three algorithms were proposed several years ago (with the latest eTRCA algorithm being proposed in 2017), this study introduced a newly proposed eTRCA algorithm improvement called TDCA for validation. TDCA is also a widely recognized training-based decoding algorithm in the field.

Furthermore, among the four aforementioned algorithms, CCA and FBCCA are both non-training algorithms, while eTRCA and TDCA are training algorithms. In this study, eTRCA and TDCA algorithms were trained on data consisting of 6 trials for each task target (stimulation frequency) using cross-validation for trial selection. A filter bank consisting of five filters was used, and its design and weight curves were kept consistent with the reference[7]. In addition, the number of components in the TDCA algorithm was selected as 9 after tuning (increasing it further did not improve performance but only added to the algorithm's complexity). Another important parameter of TDCA is Np, which refers to the length of the sliding window. Typically, the data window length used for calculation needs to be greater than 1 second, so that Np can range from 0.1 seconds to 1 second (in intervals of 0.1 seconds). In this study, when the data window length was less than 1 second, the corresponding Np parameter was set as the maximum value supported by that window length. For instance, if the window length was 0.5 seconds, Np would take values from 0.1 seconds to 0.5 seconds (in intervals of 0.1 seconds). For more detailed information regarding the principles of the four decoding algorithms mentioned above, please refer to the respective references.

2.3.1 Evaluation indexes

The core feature of SSVEP is the EEG frequency response corresponding to the stimulation frequency[34]. Therefore, the frequency signal-to-noise ratio is the most direct indicator for evaluating the strength of the SSVEP-induced response, and it has two analysis methods: narrowband SNR and broadband SNR. In addition, accuracy of recognition, optimal response time, and ITR are the core indicators of a broad consensus on the performance evaluation of the SSVEP-BCI algorithm [7, 32]. The six analyzed datasets in this study possess varying paradigm coding parameters, collection equipment, and test groups. Consequently, the results of the aforementioned five indexes typically vary significantly among different samples[9]. To summarize, this paper proposed five indexes for data evaluation, including the narrow-band signal-to-noise ratio (SNR), wide-band SNR, accuracy of recognition, optimal response time, and information transfer rate (ITR).

In addition, since different algorithms yield different results for accuracy of recognition, optimal response time, and ITR, this study needs to select a widely accepted and influential decoding algorithm in the field as a standard algorithm. FBCCA algorithm [32] is a milestone in the SSVEP-BCI field for its non-training decoding algorithm, which firstly introduced the concept of spatial filter banks and greatly improved the effectiveness of SSVEP decoding. Therefore, FBCCA is widely used in the development of BCI systems and is considered a highly influential and stable algorithm. Thus, the FBCCA was used as the standard algorithm for data analysis in this study.

The following section provides the specific definitions and calculation methods for the five indexes:

  1. The criteria of the selection of 20 most cited papers and 20 latest papers are weak. Why these papers are selected? Do they have something different to others, or do they more strongly reflect the proposed problem(performance differences when different datasets are used)?

Response:

Thank you for your suggestion! The purpose of analyzing algorithm papers here is to reflect the current situation of using 1-2 datasets for validation in SSVEP decoding algorithm research, and to support the value of the algorithm performance testing dataset system proposed in this article, and to indicate that current algorithm research can still be further expanded using more datasets. However, the paper selection and screening method here does require improvement. After careful consideration, we have made the following modifications to the content:

(1) Considering that the above content is to support the value of the algorithm performance testing dataset system proposed in this article, rather than the method and results proposed in this article, we have adjusted the content to the Discussion section.

(2) Since most of the top cited decoding algorithm papers were published a long time ago and were limited by the scarcity of publicly available datasets at that time, the analysis of their dataset usage is not meaningful. Therefore, we have deleted Table 3.

(3) We have provided more detailed explanations for the paper selection method.

(4) We have done more in-depth analysis and discussion of the content in Table 4.

After the above modifications, the final modified contents are as follows:

4.3 Current situation of dataset usage in SSVEP decoding algorithm research

This study conducted a further analysis of the current state of dataset usage in the latest research papers on SSVEP decoding algorithms. The selection of papers followed specific criteria, including using the Web of Science Core Collection with the keywords "SSVEP & algorithm" as the primary topic, limiting the document type to papers, and restricting the search results to papers published before April 2023. In addition, we manually excluded studies that were not related to decoding algorithms, such as research on BCI system applications involving algorithm validation and usage, algorithm review papers, and dataset papers. Subsequently, we sorted the search results chronologically and selected the 20 most recently published papers and analysis of their dataset usage, which is presented in Table 7.

Table 7 The current situation of dataset usage in SSVEP decoding algorithm research

Citation

Data sources

Data publicly available

Experiment

Subjects

Year

Mu et al[14]

Mu et al[40]

--

offline

9

2022

Oikonomou et al[15]

EPOC dataset[13]

Yes

offline

11

2023

Wang et al[16]

SSVEP Benchmark [8]

Yes

offline

35

2023

Guney et al[17]

SSVEP Benchmark [8]

Yes

offline

35

2023

SSVEP BETA[9]

Yes

70

Zhang et al[18]

Self-collected

--

offline

20

2023

Yin et al[19]

SSVEP Benchmark [8]

Yes

offline

35

2023

Ke et al[20]

Self-collected

--

offline

online

15

14

2023

Wong et al[35]

Chen et al[41]

Yes

offline

8 and 12

2023

Liang et al[42]

Yes

12

Chang et al[43]

--

12

Tabanfar et al[21]

Kołodziej et al[44]

--

offline

5

2023

Lee et al[22]

SSVEP Benchmark [8]

Yes

offline

35

2023

OpenBMI dataset[45]

Yes

54

Bian et al[46]

SSVEP Benchmark [8]

Yes

offline

35

2023

SSVEP BETA[9]

Yes

70

SSVEP UCSD [12]

Yes

10

Ziafati et al[23]

Zhang et al[47]

Yes

offline

10

2023

Luo et al[24]

SSVEP Benchmark [8]

Yes

offline

35

2022

SSVEP BETA[9]

Yes

70

Chuang et al[25]

Self-collected

--

offline

24

2022

Pan et al[26]

SSVEP UCSD [12]

Yes

offline

10

2022

Wang et al[48]

--

10

Oikonomou et al[27]

SSVEP Benchmark [8]

Yes

offline

35

2022

EPOC dataset[13]

Yes

11

Zhou et al[49]

SSVEP UCSD [12]

Yes

offline

10

2022

SSVEP Benchmark [8]

Yes

35

SSVEP BETA[9]

Yes

70

Yan et al[28]

SSVEP Benchmark [8]

Yes

offline

35

2022

SSVEP UCSD [12]

Yes

10

Zhang et al[29]

SSVEP Benchmark [8]

Yes

offline

35

2022

SSVEP Wearable[10]

Yes

102

Bassi et al[50]

SSVEP Benchmark [8]

Yes

offline

35

2022

SSVEP BETA[9]

Yes

70

SSVEP Wearable [10]

Yes

102

Table 7 shows that the majority of the 20 SSVEP decoding algorithm papers analyzed used more than one dataset, with only three papers [18] [25] [20] using self-collected data verification methods. Among the 20 papers, 40% (8/20) used only one dataset, while 80% (16/20) used no more than two datasets, and the remaining four papers used only three datasets for verification. Therefore, there is significant room for expansion in dataset usage in current research on SSVEP decoding algorithms.

Regarding dataset usage, it can be seen that in the 20 papers of Table 7, SSVEP Benchmark[8] (dataset1) was used 11 times, SSVEP BETA[9] (dataset2) was used 5 times, SSVEP UCSD[12] (dataset6) was used 4 times, and SSVEP Wearable[10] (dataset3) was used twice (this dataset was released in 2021). Other datasets were used no more than twice. It can be seen that there are significant differences in the frequency of use of different datasets. From the perspective of the number of subjects, it can be observed that the use of previous datasets for algorithm verification involves a much larger number of subjects than using self-collected data. Therefore, releasing SSVEP datasets can promote algorithm research. It is hoped that research teams with capabilities in this field can collect more self-generated validation data and make them public on the basis of using existing datasets.

In summary, the dataset evaluation method and algorithm testing dataset system proposed in this study can provide more reference for future algorithm research. The decoding algorithm research can attempt performance testing and verification on more newly shared datasets (such as dataset2, dataset3, and dataset4).

  1. line 260, "Based on the five SSVEP dataset evaluation indexes proposed in this paper...", isn't it 6 indexes?

Response:

Thanks for your correction. The five indexes mentioned here are obtained by feature calculation and further explained in section "2.3.1 Evaluation indexes" of the article. However, there are actually six scores as the five indexes in section 2.3.1 correspond to five scores, and the total score should also be included. We have made relevant adjustments to the article as follows:

Table 3 presents the calculation results for the six datasets based on the five SSVEP dataset evaluation indexes and six score (socre1-score5 and total score) proposed in this paper.

  1. Why FBCCA is used as standard algorithm?

Response:    

Thank you for proposing this issue. The use of FBCCA as the standard algorithm does require some explanations in the article, and relevant updates have been made as follows:

In addition, since different algorithms yield different results for accuracy of recognition, optimal response time, and ITR, this study needs to select a widely accepted and influential decoding algorithm in the field as a standard algorithm. FBCCA algorithm [32] is a milestone in the SSVEP-BCI field for its non-training decoding algorithm, which firstly introduced the concept of spatial filter banks and greatly improved the effectiveness of SSVEP decoding. Therefore, FBCCA is widely used in the development of BCI systems and is considered a highly influential and stable algorithm. Thus, the FBCCA was used as the standard algorithm for data analysis in this study.

  1. Lines 124-126, "This paper proposed six indexes for data evaluation, including the narrow-band sig-124 nal-to-noise ratio (SNR), wide-band SNR, accuracy of recognition using standard algo-125 rithm, optimal response time of standard algorithm, and information transfer rate of 126 standard algorithm." But it seems only 5 indexes.

Response:

Thanks for your correction. It should actually be 5 indexes here, and it has now been corrected as follows:

To summarize, this paper proposed five indexes for data evaluation, including the narrow-band signal-to-noise ratio (SNR), wide-band SNR, accuracy of recognition, optimal response time, and information transfer rate (ITR).

  1. Lines 290-306 are confusion. A better presentation should be considered.

Response:

Thank you very much for your suggestion! The presentation here was indeed not very clear, and has now been updated and adjusted as follows

Using a window length of 1 second, a comparison study between any two methods was conducted using the paired sample t-test. The corresponding p-value after the Bonferroni correction is listed in Table 5.

Table 5 Accuracy comparison results based on the paired t-test with Bonferroni correction

Method comparison

(M1 vs. M2)

Dataset1

Dataset2

Dataset3a

Dataset3b

Dataset4

Dataset5

Dataset6

CCA vs. FBCCA

<***

<***

N.S.

N.S.

<***

N.S.

N.S.

CCA vs. eTRCA

<***

<***

<**

<***

<***

N.S.

<***

CCA vs. TDCA

<***

<***

<***

<***

<***

N.S.

<***

FBCCA vs. eTRCA

<***

<***

<***

<***

<***

N.S.

<***

FBCCA vs. TDCA

<***

<***

<***

<***

<***

N.S.

<***

eTRCA vs. TDCA

N.S.

N.S.

N.S.

N.S.

N.S.

N.S.

N.S.

1:>***,>**,>* denote that M1 is better than M2 at different significant levels of 0.001,0.01 and 0.05.

2:<***,<**,<* denote that M1 is worse than M2 at different significant levels of 0.001,0.01 and 0.05.

3: N.S. indicates no significant difference between M1 and M2.

  1. Comments on the Quality of English Language. Few spots need to be improved.

Response:

Thanks for your question. The English expression of this article has been polished and optimized. To distinguish it from the content modifications, the adjustments made to the English language expressions are marked in blue font.

Reviewer 2 Report

This is an interesting work, it could be helpful to the scientific community expert in the topic, however I have one concern:

1) Authors review the process and also around 20 papers. However they did not follow any specific method usually applied to this type of review process. For example PRISMA guidelines or similar. The methods is the weak part of the article and it needs to be supported from a stronger approach, more systematic way. 

2) Graphs are not good. They are difficult to see. They should try another type or divide data to show in these tables. 

I consider English is ok

Author Response

Authors’ Responses

The authors thank the editor and the reviewer for their constructive comments and suggestions. We have carefully revised the manuscript according to the comments. Modifications in the manuscript have been marked in red font. The adjustments made to the English language expressions are marked in blue font in the manuscript. The detailed responses to the comments please see the attachment.

Reviewer 3 Report

Major comments:

1- In section 2.2, it is mentioned that eTRCA and TDCA are “training algorithms”. As I understand, you mean that these algorithms need training. If so, please describe how the authors trained the models of these two algorithms.  

2- Line 182, equation (5): ?????1=15.1???10(????+11): How did the authors calculate the weight “15.1” for this equation? How about the weights for equations (5) and (11)?

3- In Fig. 1, the authors showed the recognition accuracy statistics for the same algorithm on different datasets. However, they did not explain about different “data lengths”. Please describe what these data lengths are? How did you calculate them?

4- The statistical t-test results shown in lines 291-300 are very confusing. You need to demonstrate the results in a table and describe them in a clear way.

5- Lines 320-321: “The maximum value reflects the upper limit of algorithmic performance, which can be obtained by filtering operators”. This sentence is not clear. Please describe what the “filtering operators” are.

6- The English language of the manuscript needs to be improved to enhance the reader comprehension.

7- The quality of the discussion is insufficient, lacking clear definition of the main hypotheses of the study. The research questions and hypotheses should be strengthened and presented with greater clarity.

Minor comments:

8- Table 8 and related descriptions could be presented in the Results section. In the Discussion section, you may discuss it more.  

9- The authors can use bold fonts to emphasize or highlight the important values in all tables. For example, bold fonts can be used to emphasize the max and min values in each column/row of a table.

10- It’s better to move lines 136-139 to immediately after the equation (1). “Here, ?(?) refers to the signal amplitude spectrum calculated by fast Fourier transform, and ?? represents the spectral resolution. The ???? in this context is defined as the ratio of the amplitude spectrum value of ?(?) to that of the frequencies 1 Hz apart on either side, and the unit is decibel (dB)”.

11- Equations 4, 5, and 6 are used to calculate Score1, so they can be written as one equation (i.e., equation 4) with 3 conditions.  Also, this should be applied to equations 7, 8, and 9 as well as 10, 11, and 12.

The English language of the manuscript needs to be improved to enhance the reader comprehension.

Author Response

Authors’ Responses The authors thank the editor and the reviewer for their constructive comments and suggestions. We have carefully revised the manuscript according to the comments. Modifications in the manuscript have been marked in red font . The adjustments made to the English language expressions are marked in blue font in the manuscript. The detailed responses to the comments please see the attachment.    

Round 2

Reviewer 1 Report

The authors addressed all my previous comments to a satisfactory level.

Reviewer 2 Report

Authors improved the paper with the review. It is now ok for publication

English is ok

Reviewer 3 Report

I would like to extend my congrats to the authors for their efforts in enhancing the manuscript and effectively addressing the reviewers' comments.